# Trends in Preoperative Airway Assessment

**DOI:** 10.3390/diagnostics14060610

**Published:** 2024-03-13

**Authors:** Ioan Florin Marchis, Matei Florin Negrut, Cristina Maria Blebea, Mirela Crihan, Alexandru Leonard Alexa, Caius Mihai Breazu

**Affiliations:** 1Faculty of Medicine, “Iuliu Haţieganu” University of Medicine and Pharmacy, 400347 Cluj-Napoca, Romania; mateinegrut@gmail.com; 2Department of Anesthesia and Intensive Care “Clinicilor 4-6”, Cluj County Emgency Clinical Hospital, 400349 Cluj-Napoca, Romania; csbreazu@yahoo.com; 3Otorhinolaryngology Department, “Iuliu Hatieganu” University of Medicine and Pharmacy, 400347 Cluj-Napoca, Romania; cristina_blebea@yahoo.com; 41st Department of Anesthesia and Intensive Care, “Iuliu Haţieganu” University of Medicine and Pharmacy, 400347 Cluj-Napoca, Romania; crihan_mirela@yahoo.com (M.C.); alexandrualexa91@gmail.com (A.L.A.); 5Research Association in Anesthesia and Intensive Care (ACATI), 400394 Cluj-Napoca, Romania; 6Department of Anesthesia and Intensive Care, “Prof. Dr. Octavian Fodor” Regional Institute of Gastroenterology and Hepatology Cluj-Napoca, 400162 Cluj-Napoca, Romania

**Keywords:** airway assessment, difficult airway, ultrasound, computed tomography, magnetic resonance imaging, virtual endoscopy, nasoendoscopy, transnasal endoscopy, artificial intelligence

## Abstract

Airway management is a vital part of anesthesia practices, intensive care units, and emergency departments, and a proper pre-operative assessment can guide clinicians’ plans for securing an airway. Complex airway assessment has recently been at the forefront of anesthesia research, with a substantial increase in annual publications during the last 20 years. In this paper, we provide an extensive overview of the literature connected with pre-operative airway evaluation procedures, ranging from essential bedside physical examinations to advanced imaging techniques such as ultrasound (US), radiography, computed tomography (CT), and magnetic resonance imaging (MRI). We discuss transnasal endoscopy, virtual endoscopy, 3D reconstruction-based technologies, and artificial intelligence (AI) as emerging airway evaluation techniques. The management of distorted upper airways associated with head and neck pathology can be challenging due to the intricate anatomy. We present and discuss the role of recent technological advancements in recognizing difficult airways and assisting clinical decision making while highlighting current limitations and pinpointing future research directions.

## 1. Introduction

Despite certain technical advances in the field, difficulties or failures in securing the airways still occur and are related to severe morbidity and mortality [1]. Patients with or without a suspected problematic airway may pose significant challenges if the preprocedural airway assessment fails to provide an objective picture leading to proper airway management.

Complex airway assessment has recently been at the forefront of anesthesia research, with a substantial increase in annual publications during the last 20 years [2].

Preoperative airway evaluation is an important aspect that an anesthesiologist considers for general anesthesia patients. The purpose of an airway assessment in preparation for airway management, independent of the context, is to identify and predict potential problems related to maintaining oxygenation during a procedure and draw up a fallback plan in the event of an unanticipated problematic airway. A preoperative airway evaluation comprises tests and investigations that should ideally be simple and cost-effective, with high specificity and sensitivity and a high positive predictive value. However, the diagnostic accuracy of different tests varies widely. A traditional preoperative bedside examination includes tests that lack the sensitivity to predict difficult facemask ventilation, laryngoscopy, and tracheal intubation [3,4,5,6,7]. Solid evidence suggests deficiencies in planning across all fields that interact with airway management [8]. A closed-claims analysis of difficult tracheal intubation found that three-quarters of the cases involved judgment failures resulting in severe complications, including death and hypoxic brain damage [9].

The gold standard of a preoperative assessment of the airway is not well defined, but a comprehensive evaluation should typically include the following:A thorough bedside objective examination based on the current guidelines’ recommendations;Obtaining a history of prior airway difficulties;Looking for specific preexisting acquired or congenital medical conditions and prior treatments that are likely to complicate airway management;Conducting an advanced examination that includes an endoscopy or imagistic tests if there is a problematic airway suspicion;Evaluating access to the cricothyroid membrane [10].

The overall incidence of unsuccessful intubation or CICO (cannot intubate, cannot oxygenate) situations in anesthesia practice is low, with death or permanent brain injury occurring in 1 in 180,000 anesthetics, with an estimated 25% of fatal airway incidents reported as such [8]. A third of the severe airway complications reported by the Fourth National Audit Project of the Royal College of Anesthetists and the Difficult Airway Society (NAP4) were in patients with head and neck pathology. For most of them, the anesthesiologist failed to assess the airway correctly, resulting in poor management [8]. An unanticipatedly difficult airway can result in dire consequences such as death, hypoxic brain injury, cardio-respiratory arrest, aspiration pneumonia, trauma to the vocal cords, and dental injuries. In an emergent, non-operating room setting, such hazardous outcomes can be even more frequent, which might be due to a lack of proper patient positioning, available equipment, and skilled physicians, as well as the inability to perform a proper airway assessment or adequate physiological status optimization before intubation [11,12,13,14].

These complications justify developing and implementing reliable airway assessment tests, guidelines, and algorithms, emphasizing the need for training in difficult airway identification and evaluation using daily practice tools like nasal endoscopy or ultrasonography.

This review aims to evaluate advanced preoperative airway assessment techniques and discuss their applications as routine complementary tests in high-risk patients.

We intend to provide a comprehensive analysis of the arising tools for airway assessments in addition to traditional bedside evaluations.

Since head and neck pathology significantly contributes to airway difficulty and poor patient outcomes, we will focus on this topic when presenting the advances in airway assessments.

## 2. Methodology

We searched the PubMed, Embase, and Scopus databases from 8 January 2024 to 26 January 2024, including studies published starting with the year 2000, using a combination of the following keywords: “airway evaluation”, “airway assessment”, “airway clinical exam OR airway physical exam OR airway bedside exam”, “upper airway ultrasound OR upper airway sonography”, “computed tomography”, “virtual endoscopy”, “magnetic resonance imaging”, “flexible laryngoscopy”, “artificial intelligence”, “preoperative airway assessment”, “difficult intubation”, “difficult laryngoscopy”, and “difficult airway”. A summary of our search strategy is shown in Figure 1. We prioritized the most recent meta-analyses, systematic reviews, randomized controlled trials, representative associations’ and focus groups’ guideline statements, and studies with larger sample sizes for inclusion. The exclusion criteria were non-human studies, conference proceedings, editorials, and articles not written in English. We included studies involving patients with head and neck obstructive pathology matching our search interest.

After duplicate removal, titles and abstracts were manually screened against our exclusion criteria. Full-text articles were then analyzed, and we reached the final list of included studies by consensus. Prospective observational trials written in the last five years, with a larger sample size, were preferentially selected, but older, retrospective studies were also included when no other data were available. Given the significant heterogeneity of the literature and the broad scope of our paper, no statistical analysis was performed. Instead, we opted for a narrative review, highlighting recent advancements and trends in perioperative airway assessment, current limitations, and future study directions.

## 3. Results

### 3.1. Physical Examination

Anticipating a problematic airway allows one to adopt prophylactic measures and provides valuable time for adequate preparation. Various predictive factors have been developed, along with specialized medical care, aimed at indicating the risk of a difficult airway, comprising demographic information, medical histories or records, and physical examination findings [15,16,17].

However, although these factors, including bedside examinations and tests, may facilitate the identification of difficult airways, they lack the statistical power for safe practice in a busier and more complex operating theater, intensive care unit, or emergency department [17].

The Mallampati score is an assessment to describe the relative size of the tongue base compared to the oropharyngeal opening to predict the difficulty of an airway. The Mallampati score is a basic test that is carried out when a patient requires anesthesia or must be intubated, allowing the anesthesiologist to prepare for challenges related to airway anatomy prior to securing the airway [16,18,19].

The modified Mallampati score used routinely today is as follows [17,18]:Class 0: any part of the epiglottis is visible.Class I: soft palate, uvula, and pillars are visible.Class II: soft palate and uvula are visible.Class III: only the soft palate and base of the uvula are visible.Class IV: only the hard palate is visible.

Numerous studies have demonstrated bedside tests’ reduced positive predictive value if they are evaluated individually. At the same time, their absence is a relatively accurate indicator of a high probability of maintaining airway control [17,18,19]. However, combining these parameters by developing risk scores increases their sensitivity, justifying their use in airway evaluation. The Wilson test is such a multivariate score that combines five parameters (Table 1) [20,21].

Of the traditional bedside tests, the upper lip bite test (inability to bite the upper lip with the lower incisors) arose as a bedside test that practitioners can efficiently perform to predict difficult airways, showing a higher sensitivity, although still modest [17].

The upper lip bite test, which involves instructing patients to bite their upper lip with their lower incisors, offers a classification system based on the following criteria [15,16,18,19]:Class 1: the lower incisors extend beyond the vermilion border of the upper lip;Class 2: the lower incisors can bite the lip but cannot extend above the vermilion border;Class 3: the lower incisors cannot bite the upper lip.

A complete airway evaluation must include an assessment of the predicted ease or difficulty of orotracheal intubation and the predicted success of backup options to maintain oxygenation, such as face mask ventilation, supraglottic device use, and a surgical airway [18,19]. As the number of predictors of difficulty increases, the probability of encountering a problematic airway is also higher.

Mask ventilation is an essential component of airway management and general anesthesia administration. Successful mask ventilation provides anesthetists with a rescue technique during unsuccessful laryngoscopy attempts and in unexpectedly difficult airway situations (Table 2) [19,20,21]. The most critical situation is when intubation is impossible and mask ventilation is inadequate or becomes inadequate. Consequently, anticipation of difficult mask ventilation is of vital importance [21,22,23].

With an anticipatedly difficult tracheal intubation, an assessment of the likely success of backup oxygenation with a face mask or the use of a supraglottic device is particularly justified. Predictors of difficult ventilation using a supraglottic device are present in the literature [23,24,25]. In most situations, a significant difficulty predicted for both tracheal intubation and ventilation with a face mask or a supraglottic device should be considered a strong signal to consider awake intubation, especially for a cooperative patient undergoing elective surgery [24,25]. The features associated with poor supraglottic device compliance are as follows:Reduced mouth opening;Supra- or extraglottic pathology (neck radiation, tumors, cysts, lingual tonsillar hypertrophy);Glottic and subglottic pathology;Poor dentition;Male sex;Applied cricoid pressure;Obesity;Fixed cervical spine flexion deformity;Rotation of the head.

The laryngeal mask (LMA) represents significant progress in airway management and has been included in difficult airway management algorithms. The LMA has been used successfully in patients with difficult airways, including patients in whom fiberoptic intubation (FIB) has failed. Therefore, the LMA may be a valuable alternative to FIB in complex airway management [23,24].

Various congenital and acquired conditions have been associated with difficult airways (Table 3) [26]. In addition to these conditions, pneumonia, gastric content aspiration, asthma, and chronic obstructive pulmonary disease may alter oxygenation and lung ventilation, further complicating airway management.

Neck and head pathology significantly contributes to acquired difficulties of the airway, some of which are vital emergencies. NAP4 acknowledges the deficiencies connected with airway management in ENT (ear, nose, and throat) patients and recommends imagistic or endoscopic exams when such pathology is suspected [3,17]. Therefore, screening for potential disorders that distort the airway and impede airway management in an unconscious patient is vital. A first-glance assessment should check voice quality, signs of upper airway obstruction, and cervical masses. Changes in speech or voice, like hoarseness or a “hot potato” voice, should alert the practitioner. Latero-cervical adenopathy, a history of airway surgery or radiation, trismus, stridor, or obstructive sleep apnea are signs of a potentially problematic airway, and further investigations are mandatory. Identifying the cricothyroid membrane prior to airway management through a physical exam or ultrasound could save precious time in a critical situation [27].

Tracheal pathology, which may include tracheal stenosis with various locations, a tracheo-esophageal fistula, extrinsic compression, or patients with a history of stented trachea who need surgery for miscellaneous reasons or present with post-stenting complications, poses a significant anesthetic challenge mainly due to problematic airway management [8,10]. A fixed obstruction of the subglottic area or upper trachea mainly causes inspiratory symptoms, while obstructive pathology involving the thoracic trachea exhibits expiratory symptoms. As the clinical signs are discrete until advanced obstruction grades, careful anamnesis of a patient’s history and auscultation of airway flow on a tracheal level may guide a practitioner to more complex investigations. The vital steps are determining the location and length of the stenotic lesion and the position that allows the patient to breathe most comfortably, assessing the feasibility of mask ventilation or supraglottic device use, and developing complex airway management strategies [8].

### 3.2. Nasoendoscopy

#### 3.2.1. Introduction

The current guidelines highlight the importance of a preoperative bedside endoscopy of the upper airway as an aid for the practitioner to elaborate a safe airway management plan [10]. A transnasal flexible endoscopy is an established diagnostic tool for patients with hoarseness, stridor, or suspected pharyngo-laryngeal cancer [28]. These patients generally need general anesthesia so that they will benefit from the images stored during the endoscopic exam. However, convincing evidence on the usefulness of a preoperative endoscopy in the general population requiring general anesthesia is lacking. Still, there has been increasing interest in the matter as fiberoptic technology has become available, and a few studies have shown that transnasal fiberoptic airway assessments could improve the predictability of routine airway examinations regarding conventional laryngoscopy or videolaryngoscopy [29,30,31].

#### 3.2.2. Procedure

Transnasal flexible endoscopic laryngoscopy (TFEL) is a minimally invasive procedure that involves using a fiberscope with a 300–350 mm working length and a 2.5–3.5 mm diameter (flexible rhinolaryngoscope) to inspect the nasal cavity, pharynx, and larynx (Figure 2). Because of its smaller size, this type of fiberscope is safe, minimally invasive, and handy. It advances through one of the nares below the inferior turbinate until it reaches the pharynx, with the patient in a semi-recumbent or sitting position [4]. The tongue base and the epiglottis are usually visible at this point unless there is obstructing pathology like tumors, inflammation, or anatomical anomalies impeding visualization. Sniffing, protruding the tongue, vocalizing, or taking deep breaths improve the visualization of structures of interest. ENT specialists have adopted the tool in their daily practice, routinely performing this procedure in outpatient settings or emergencies. Still, other professionals (anesthetists, emergency medicine specialists, or pulmonologists) can use them and quickly pick up the expertise, as the learning curve is fast. Patients rate the procedure’s tolerability as excellent; cooperative children and even newborns cope with the maneuver very well; and serious complications are rare [32]. While inspecting the upper airway, a professional can visualize the anatomical and pathological structures, assess the pharynx’s or larynx’s functionality, or even perform interventions in the inspected area [33]. To mitigate the patient’s discomfort or the risk of bleeding, vasoconstrictors and topical anesthesia are applied to both nostrils. Nebulized or dispensed over the mucosa with an atomizer, 2–4% lidocaine should supply adequate upper airway analgesia during the procedure if the exam needs to cover the subglottic area.

#### 3.2.3. Indications

The main indications of a flexible rhinolaryngoscopy are the following:Persistent hoarseness or stridor;Tumors or infections;Allergic reactions with possible airway involvement;Pre- or post-operative assessment;Evaluation of obstructive sleep apnea;Deglutition;Inspection for a foreign body;Acute airway exam: trauma, stridor with respiratory failure, or oropharyngeal bleeding;Minor interventions.

In an ambulatory or hospital elective setting, a nasendoscopy provides an objective assessment of the airway and warns about the presence of pathological pharyngolaryngeal structures, their location, and their potential impact on airway management.

A standard bedside airway assessment does not address the pathology of the tongue, epiglottis, glottic opening, or larynx, as symptoms related to this pathology are often discrete. However, they pose a significant challenge during airway management. Rosenblatt et al. [34] demonstrated that using preoperative FNE changed the planned airway management in 26% of patients, either opting for awake intubation or giving up the initial plan to perform awake intubation. A preoperative flexible nasoendoscopy with tongue protrusion has also been shown to improve the prediction of the modified Cormack–Lehane grade during a conventional laryngoscopy [31]. A prospective feasibility study and a case series report showed that a fiberoptic nasal endoscope armed with an intubating tube could be used for the awake nasotracheal intubation of a cohort with upper airway pathology, revealing the possibility of using this device for both airway evaluation and management in elective cases or emergencies [35,36].

Examining an airway emergency through the use of a flexible rhinolaryngoscope may prove essential for prompt and life-saving management, as other exams may be time-consuming and jeopardize an already compromised airway (Figure 3). Nasoendoscopy offers a fast and clear picture, and it could be the best choice for a patient with stridor and upper airway respiratory failure of known or unknown cause presenting in the emergency department to establish if patients require intensive care unit (ICU) admission or urgent airway management [37]. Patients with a history of prolonged intubation or tracheostomy may suffer from different grades of tracheal stenosis, often oligosymptomatic, and greatly benefit from a preoperative endoscopic evaluation under topical anesthesia that includes the central airway. Critical and complex stenosis located distally on the trachea poses a significant challenge regarding airway management. Precipitous surgical access to the trachea could lead to disastrous consequences in such cases [8]. Patients with obstructive sleep apnea can be assessed through flexible nasopharyngoscopy for surgery indication [38].

A summary of studies regarding nasoendoscopy and difficult airway prediction is presented in Table 4.

#### 3.2.4. Limitations

The image obtained through a flexible laryngoscopy differs from the view provided by a conventional direct rigid laryngoscopy or videolaryngoscopy, as the airway approach is different [4]. The position of the patient and the pharyngeal tone may further contribute to this, so when devising an airway management plan, a practitioner should consider these issues. However, some studies have found strong correlations between transnasal fiberoptic glottic examination and videolaryngoscopic performance [30,31,32]. Complications occur rarely and may include epistaxis, discomfort, and exceptional laryngospasm. When performed in an emergency, added care should be employed, as even a little injury over an already compromised pathological airway could lead to severe edema or bleeding.

### 3.3. Ultrasound

#### 3.3.1. Introduction

Ultrasound imaging is a noninvasive, simple, and portable tool that may help clinicians with rapid airway assessment and management in operating rooms, emergency departments, and intensive care units [41]. Point-of-care ultrasound (POCUS) has emerged as a valuable diagnostic and therapeutic aid for a wide range of procedures, with many studies highlighting its promising role in airway assessment over the last decade [42,43,44,45].

There are various clinical uses for POCUS in airway management, starting with assessing fasting status before airway management. Ultrasound can check the proper endotracheal tube placement (ETT), which comes in handy in scenarios where end-tidal capnography is not available or reliable. However, it can also identify ETT malposition, such as mainstem intubation or esophageal intubation. Another clinical application is offering guidance for percutaneous tracheostomy (depth and the presence of any overlying vessels at the proposed insertion site) or identifying the cricothyroid membrane (CTM), preparing attending staff for an emergency cricothyroidotomy in a cannot intubate, cannot ventilate scenario. Also, users may detect subglottic stenosis, predict difficult intubation, detect readiness for extubation (absence of glottic edema), and predict pediatric ETT and double-lumen tube (DLT) size [41,42,43].

To obtain quality US scans of airway anatomy, users require basic knowledge of ultrasound physics, proper transducer selection, and proper probe orientation [41].

#### 3.3.2. Sonoanatomy of the Upper Airway

Even though many anatomical structures in the upper airway compartment contain air, with the help of ultrasound, one can recognize vital upper airway structures like the tongue, epiglottis, hyoid bone, vocal cords, thyroid cartilage, cricothyroid membrane, cricoid cartilage, and tracheal rings [44].

Five major ultrasonographic views are used in assessing the upper airway: the suprahyoid, thyrohyoid, thyroid, cricothyroid, and suprasternal views [42]. Each view assists users in identifying and confirming the prominent landmarks of the upper airway POCUS (Figure 4).

The suprahyoid view provides valuable information about tongue thickness and can be used to measure the hyomental distance (HMD) (Figure 5) [45]. The epiglottis appears from the thyrohyoid view as a thin stripe anterior to a hyperechogenic air–mucosa interface; the entire image can be associated with a small face sign [46]. Over the thyroid cartilage, three main structures come into view: vocal cords, strap muscles, and arytenoid cartilage (Figure 5) [47].

The cricothyroid view lies between the cricoid cartilage and thyroid cartilage, with its leading practical utility being to locate the cricothyroid membrane, which is perforated during a cricothyrotomy (Figure 5). Another way to identify the cricothyroid membrane is using the TACA protocol (thyroid–airline–cricoid–airline) with the probe in the transverse plane [48]. By ultrasound scanning in the sagittal plane, the thyroid cartilage can be visualized from a superficial to deep depth as the sample moves toward the cricothyroid membrane. The cricoid cartilage, with a hypoechoic ovoid structure, will appear inferior to the cricothyroid membrane. Finally, the tracheal rings show some small, well-spaced, intermittent hypodensities. This sagittal image is called “beads on a string” [48]. Placing the transducer at the suprasternal notch for a transverse view facilitates the tracheal visualization as an inverted U-shaped structure with a smooth mucosal lining marked by an air–mucosa interface. By evaluating the transverse tracheal diameter and mucosal irregularity or loss of air–mucosa interface, the US may prove to be a good tool for bedside diagnosis of post-intubation tracheal stenosis [44].

#### 3.3.3. Ultrasound Assessment in a Difficult Airway

The most common method of evaluating a potentially difficult intubation scenario with the help of ultrasound is an assessment of the hyomental distance. This represents the distance between the hyoid bone’s upper edge and the chin’s lower edge. In the case of a direct laryngoscopy, the hyomental distance ratio can also be used, which is the distance measured with the head in a neutral position and with the cranial extremity in the hyperextension position, measured, of course, by ultrasonography. The most recent data showed that if the hyomental distance is ≤1.09 cm, then intubation will be difficult [42,49,50]. Table 5 summarizes the most relevant recent studies regarding ultrasound evaluation of the airway.

### 3.4. Radiographic Evaluation

Anteroposterior or lateral radiography of the trachea is relatively easy to obtain during routine chest examinations. Possible changes in size, foreign bodies, deviations from the midline, and radiological confirmation of the correct or incorrect positioning of the intubation tube can be detected [68]. In the case of planning a selective intubation of the lungs with a double-lumen probe, the angle of tracheal bifurcation and the diameter of the main bronchi are analyzed, and the intubating tube is chosen accordingly [68]. Along with this routine examination, craniocervical X-rays dedicated to airway examinations are helpful in pathological situations with complex airway potential: trauma, acromegaly, genetic syndromes, and degenerative diseases [68,69]. Several studies have focused on the association between difficult airways and specific osteoarticular or anthropomorphic changes observed during a cervicofacial radiological examination. Poor visualization of the glottis during a classical laryngoscopy has been associated with changes in the shape and size of the mandible, degenerative processes of the atlantoaxial or atlantooccipital joint in the context of general pathology, and a maxillopharyngeal angle of less than 90° [68,70,71]. Radiographic images should be reviewed by anesthetists whenever available, but given the low sensitivity and specificity of radiography-derived parameters, it is questionable whether patients should be exposed to unnecessary irradiation for an airway evaluation alone [72,73]. Radiography might be particularly useful in places where ultrasound is not accessible or there is no proficiency in its use. Table 6 summarizes the main papers regarding radiography and difficult airway prediction.

### 3.5. Computer Tomography, Magnetic Resonance, and Virtual Endoscopy

Anesthetists and ENT surgeons share responsibility for the airway, so it is critical that they communicate the airway strategy and carry out a thorough preoperative examination together. Patients with head and neck cancer face particular difficulties with their airways due to tumors of the oropharynx, larynx, and hypopharynx that significantly constrict and deform the airway’s structure.

In order to help with treatment planning and as part of the disease staging protocol, the majority of patients with head and neck diseases undergo cross-sectional imaging, such as computed tomography (CT), cone beam (CB) CT, or magnetic resonance imaging (MRI). Analyzing these pictures will usually yield comprehensive details on the tumor’s anatomical location, its involvement with surrounding structures, and whether the airway is distorted or narrowed (Figure 6 and Figure 7).

Table 7 summarizes relevant studies regarding CT parameters and their predictive value for difficult airways.

These two-dimensional static scans do not permit dynamic intraluminal assessment or well-informed airway planning. Virtual endoscopy (VE) has been rapidly emerging in recent years as a useful tool that fills in the gaps in airway assessment by offering a noninvasive, anatomically accurate picture of the intraluminal geography of the airway, including the supraglottic, glottic, and subglottic structures. A flexible endoscopic central airway assessment is the first choice for the fast evaluation and staging of tracheal stenosis due to its availability and cost. However, a radiological evaluation, such as helical CT with virtual endoscopy, is a prerequisite for the scoring system, planning complex airway management, and choosing between surgical or endoscopic treatment options. In a study by Kariem El-Boghdadly et al. [84], anesthesiologists were asked questions regarding the diagnosis of airway pathology and airway management strategies. When VE was added, the diagnostic accuracy rose from 54.1% with CT alone to 67.7%. The addition of VE to the clinical history and CT resulted in modifications to the airway management strategy in 48% of instances, with 90.6% of these modifications being classified as more conservative.

The head and neck have intricate anatomies where many vital organs, arteries, and nerves come together, resulting in variations in tumor forms and growth patterns. Better preoperative three-dimensional planning is now possible thanks to the advancement of digital technologies [85]. Precision medicine in head and neck surgery is now feasible thanks to the quick application and combination of virtual surgery (virtual reality and augmented reality), CAD (computer-aided design), CAM (computer-aided manufacturing), rapid prototyping (RP), 3D navigation, and robotics in a personalized manner [86].

Viewing CT digital imaging processed with a 3D modeling software (D2P; 3D Systems, Inc., Rock Hill, SC, USA) using a VR system is the next step for up-close visualization and the ability to change or combine views when assessing the airway. The views obtained through the airway are similar to a combination of those obtained using flexible bronchoscopy and direct or videolaryngoscopy [87,88].

Virtual endoscopy, although used by radiologists for many years, is a new concept in surgery and anesthesia. It involves importing patient images into three-dimensional (3D) software [85,89]. The software allows for the generation of an accurate anatomical representation of the complete airway in a 3D representation. Rapid prototyping is a flexible technology that is widely used in industry [90,91]. Such virtual visualization of the airway facilitates a bespoke airway strategy based on the patient’s preexisting diagnostic CT images. Subsequent recording of the real-time intubation allows for a retrospective comparison with the virtual endoscopy to assess its fidelity. This process facilitates the provision of a safe airway and offers opportunities for safe training through virtual rehearsal, manikin intubation, and a review of complex airway cases [88,90,92].

We need to consider that some limitations to CT scans are present with 3D VR reconstruction. When using CT scans to support complex airway management planning, it is essential to remember that the scans are static images of the patient’s airway obtained while the patient is supine.

### 3.6. Artificial Intelligence

In recent years, artificial intelligence (AI) has become an ever-growing area of research in anesthesia practice, with multiple clinical applications, from aiding in peripheral nerve blocks to predicting hypotension [93] and, more recently, guiding airway management.

Using facial structure analysis software and patient photographs, Connor et al. [94] generated a mathematical model of a patient’s head and neck. Of the 61 facial proportions analyzed, 11 were able to discriminate between easy and difficult intubation, and three facial parameters (the brow–nose chin ratio, nose tilt, and jaw–neck slope), together with the TMD, yielded the final model, with a sensitivity, specificity, and AUC of 90%, 85%, and 0.899, respectively.

Cuendet et al. [95] created a fully automated system using photobooth-like equipment that collected patient photos and videos while they made certain facial and head motions. The automatic parametrization of facial features and a random forest algorithm were used, and the resulting model was able to classify patients into easy, intermediate, and difficult intubation groups (Se = 72.9; Sp = 68.4; AUC = 77.9).

Hayasaka et al. [96] created the first convolutional neural network (CNN) AI model that predicts intubation difficulty based on a patient’s external facial features. They used side pictures of supine patients in a neutral neck position with their mouths closed to create a model with a good predictive capacity (Se = 80.5%; Sp = 83.3%; and AUC = 0.864). They implemented class activation heat maps to investigate which focus points were significant in predicting airway difficulty. Viewpoints were concentrated along the chin, tip, and neck area in the images classified as easy intubation. In contrast, in the complex intubation group, the viewpoints were dissipated, suggesting that a wide range of external anatomical features can influence the predictability of airway difficulty.

Wang et al. [97] used a semi-supervised deep-learning method to predict difficult airways. When labeling only 30% of the training sample as easy or difficult intubation, the AI model had a good performance for predicting airway difficulty (Se = 89.58%; Sp = 90.13%; AUC = 0.9457), very similar to that obtained when using a fully supervised learning scheme. This novel semi-supervised approach requires less time and resources and is more cost-effective while maintaining adequate accuracy.

Using a neural network, Xia et al. [98] created a model based on external facial features only (from pictures in the head-up, tongue extension, mouth open, and lateral positions) that predicted difficult videolaryngoscopy with a sensitivity, specificity, and AUC of 75.7%, 72.1%, and 0.779, respectively. This model had greater accuracy than alternative models, including facial features, baseline characteristics, medical history, and bedside examinations.

Pei et al. [99] used 3D face scanning technology and machine learning to predict difficult mask ventilation; the resulting model has a sensitivity, specificity, and AUC of 82.9%, 73.3%, and 0.825, respectively.

## 4. Case Scenarios

### 4.1. Case Scenario 1

A 58-year-old male presented to an emergency surgery department with a diagnosis of acute cholecystitis. He was proposed for an emergency laparoscopic cholecystectomy. During the pre-anesthetic exam, he complained of minor discomfort when swallowing, and his voice seemed distorted and muffled. Otherwise, he seemed fit without remarkable comorbidities. The anesthetist decided to perform a preoperative nasal endoscopy. The examination revealed a large mass covering the epiglottis and vallecula that would have obstructed a direct laryngoscopy or rigid videolaryngoscopy (Figure 8). After topicalization and light sedation, he performed awake nasotracheal intubation using a fiberscope with a 6.5 mm reinforced intubating tube. The procedure was fast, well tolerated, and uneventful. The laparoscopic cholecystectomy took place without any incidents, and the patient was referred to an ENT service for further investigations.

### 4.2. Case Scenario 2

A 53-year-old female patient with persistent stridor for two weeks presented at our hospital and was referred from another institution. She had grade 1 obesity, and she was diagnosed with asthma about two years ago and treated with inhaled medication. She was a moderate smoker without any other known comorbidities. A CT scan a week before admission revealed an expansive subglottic mass occupying the laryngeal lumen with a drastically reduced respiratory space and no adenopathy (Figure 9). Upon admission, the patient was in obvious distress, with severe stridor and an oxygen saturation of 94–95% with supplemental oxygen via a face mask. She received 200 mg of hydrocortisone intravenously in the ambulance. The patient categorically refused the option of a tracheostomy. The physical exam revealed a Mallampati grade 3 with a 3 cm inter-incisor distance at maximum mouth opening and a short neck. After we administered aerosolized lidocaine 2% + epinephrine 1/20,000, the stridor diminished, and we investigated the patient with a transnasal video rhinolaryngoscope, which confirmed the CT scan findings (Figure 9). We informed the patient that we could try awake fiberoptic intubation, but with the option of an emergency tracheostomy. We continued topical anesthesia with instilled lidocaine 2% through a pediatric fiberscope without sedation. We performed an airway ultrasound to identify the trachea and its depth and mark the incision point in case of an emergency tracheostomy. Because of the tumoral location, a cricothyroidotomy did not seem feasible. We tried and succeeded in intubating the patient in a sitting position with a 5.5 mm ID tube via the nasal route. The surgeon performed a tumoral debulking with biopsy and hemostasis, and we extubated the patient shortly after surgery.

## 5. Discussion

As time is of the essence during airway management, it is vital to carry out a proper airway assessment and plan accordingly before a procedure. The importance of this step may come to light, although some other essential factors also play a role when comparing the rate of complications associated with airway management in an operatory room and outside of it, e.g., intensive care units, emergency departments, or out of hospital [14,101].

A physical evaluation comprising the traditional bedside tests is still essential in selecting high-risk patients and acts more accurately as an exclusion tool. Tests like the “upper lip bite” or Wilson score are at hand and provide a valid first-glance picture [19].

A fiberoptic laryngoscopy allows for a comprehensive and detailed examination of the upper airway, including the vocal cords and subglottic area. It is a fundamental tool with daily application in every ENT office. Flexible rhinolaryngoscopy is not used to its full potential in anesthesia, intensive care, or emergency departments, despite recommendations in guidelines to consider it in patients with possible airway pathology of a tumoral, inflammatory, or traumatic nature that could preclude mask ventilation and impede the placement of a supraglottic device or an intubating tube [10,28]. Its utility in assessing anatomically difficult airways is yet to be determined, but a few studies have shown its benefit [29,30]. As technology progresses, this device will become handier, more affordable, and provide sharper images, and patients rate it as having good procedure tolerability. In this context, there is every premise that the flexible rhinolaryngoscope will soon find a place as a routine in the armamentarium of anesthetists or emergency medicine doctors for airway assessment and as an aid for airway control in elective or emergent circumstances.

The videolaryngoscope has drawn accolades as the preferred tool for difficult intubation, and it proved feasible for awake intubation with a performance comparable to fiberoptic intubation [102,103]. Although the rigid videolaryngoscope has become a core component of modern airway management, its place in airway assessment is yet to be determined since the literature is scarce. Nevertheless, as the videolaryngoscope offers good glottic visualization during awake intubation, it is logical to assume that it could prove helpful in airway inspection and guide a correct strategy for airway control [104]. The videolaryngoscope has become widely available, and there are reports of its use for an “awake look” under topical anesthesia before choosing a particular technique for airway management [105]. The “awake look” seems to have great practicability before building an awake or anesthetized airway management plan. However, caution should be expressed due to the limitations of the procedure in patients with an active gag reflex, limited mouth opening, or large friable masses of the hypopharynx and tongue base [106,107].

POCUS is a simple, safe, and valuable tool that helps clinicians confirm endotracheal intubation, measure the anatomical distances of the upper airway, and practice invasive procedures. Upper airway ultrasound is practical in airway management because of its accessibility and lack of invasiveness. A limitation of ultrasound imaging is that it only visualizes the anterior structures, and the results are operator-dependent.

Although airway ultrasound was extensively studied to predict difficult intubation, the results are heterogeneous, and it is unclear what parameters should be measured and what the cut-off values are for difficult laryngoscopy prediction. A meta-analysis by Carsetti et al. in 2022 concluded a weak positive predictive power of ultrasound in this matter [108]. However, the role of ultrasound for cricothyroid membrane identification, laryngeal edema identification, the confirmation of tracheal intubation, or the identification of the trachea prior to a tracheostomy is beyond doubt [42].

The management of a distorted upper airway associated with head and neck pathology can be challenging due to its intricate anatomy. Traditional instruction typically entails books and cadaveric dissection, which takes a lot of time and effort [109]. An evaluation of the airway using fiberoptic laryngoscopy, CT, and MRI has brought about consistent improvements in determining the management of difficult airway cases. However, new developments in 3D reconstructions—such as volume rendering, virtual reality, augmented reality, cinematic rendering, 3D modeling, and 3D printing—are being used more and more to plan complex cases, offering fresh perspectives on difficult-to-operate head and neck tumors [110].

In the future, mobile phone apps might assist airway specialists in making clinical decisions. Aguilar et al. [111] developed an app using convolutional neural networking to predict difficult intubation based on the Mallampati score, classifying patients into low-risk (Mallampati 1–2) or high-risk (Mallampati 3–4) patients with an 88.5% accuracy. While this might be of little importance for experienced anesthetists, more complex mobile apps that aim to predict difficult intubation based on external facial features are starting to develop. This might be useful in a pre-hospital setting, where paramedics could quickly assess the risk for a difficult airway using such apps so that adequate equipment and skilled staff are made available before the patient deteriorates.

AI prediction technologies are still in their early days; most studies are small, single-center studies and do not account for geographical and ethnic differences in patient populations. Developing an AI algorithm that predicts difficult airways based on pre-existing imaging (such as head and neck CT or MRI) could guide clinical decision making and improve patient safety.

## 6. Conclusions

As the need for complex airway management inside and outside of operating rooms constantly increases, so does the burden on professionals involved in this field. Airway incidents pose significant stress for healthcare staff, and their potential outcomes may lead to severe consequences. Our paper, inspired by the literature and experience, suggests that the means needed to improve the quality of airway assessment are accessible and easy to use. As new tools for airway control have been developed, diversified, and widely adopted in recent decades, there is a need to proceed with airway evaluation beyond traditional bedside examinations.

## Figures and Tables

**Figure 1 diagnostics-14-00610-f001:**
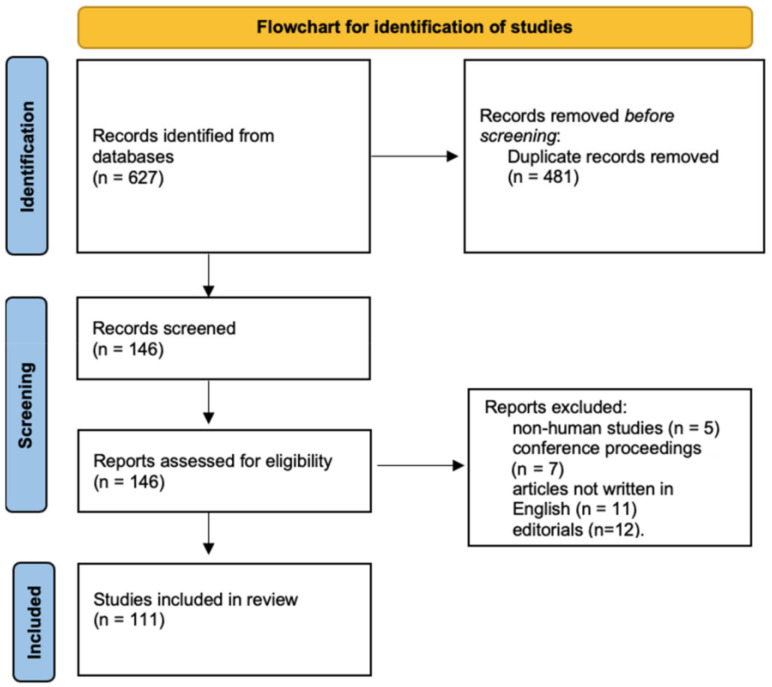
Flowchart for study selection.

**Figure 2 diagnostics-14-00610-f002:**
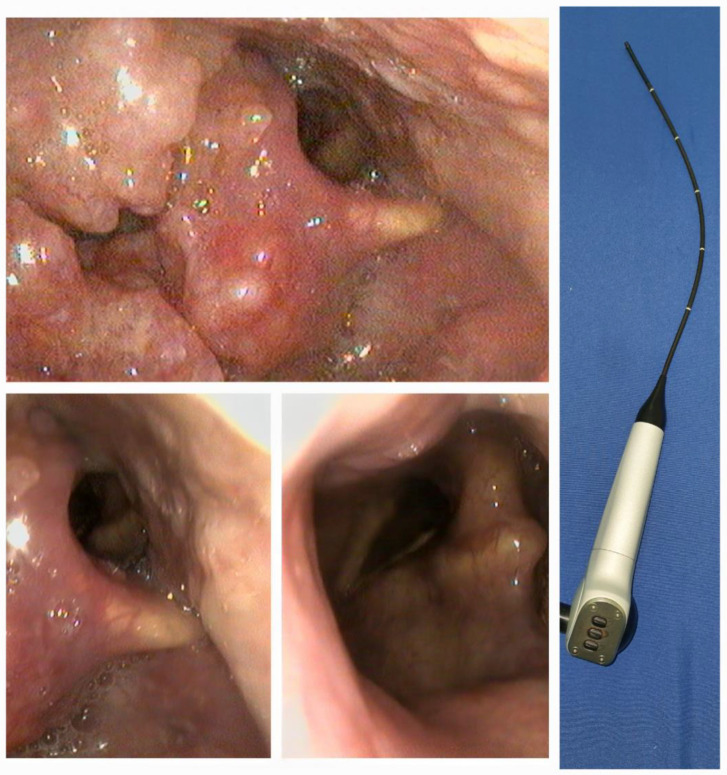
Endoscopic image of a tongue base tumor with epiglottic invasion (**left**). Flexible rhinolaryngoscope (**right**).

**Figure 3 diagnostics-14-00610-f003:**
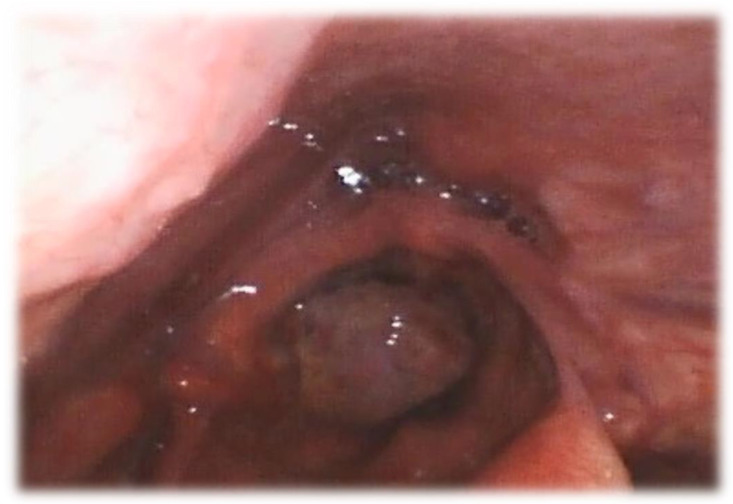
Nasoendocopic view of a supraglottic tumor with an extensive glottic obstruction.

**Figure 4 diagnostics-14-00610-f004:**
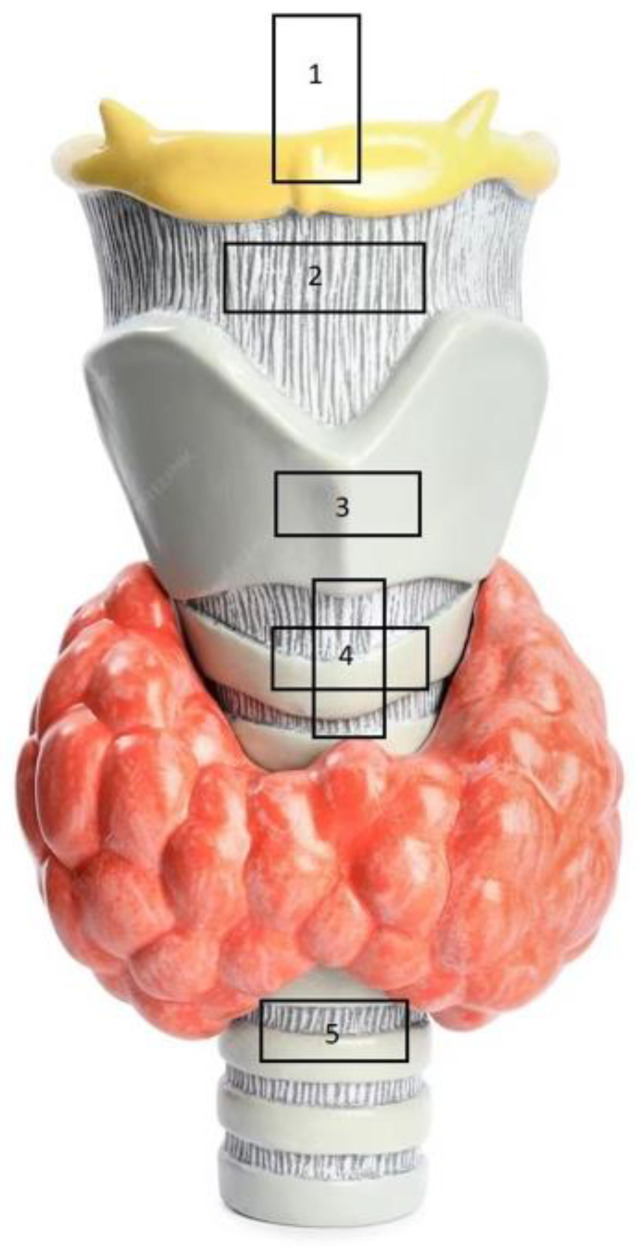
POCUS views of upper airway. 1: suprahyoid (assesses oral space). 2: thyrohyoid (epiglottis identification). 3: thyroid (status of vocal cords). 4: cricothyroid (identification of the cricothyroid membrane). 5: suprasternal (confirmation of endotracheal tube placement).

**Figure 5 diagnostics-14-00610-f005:**
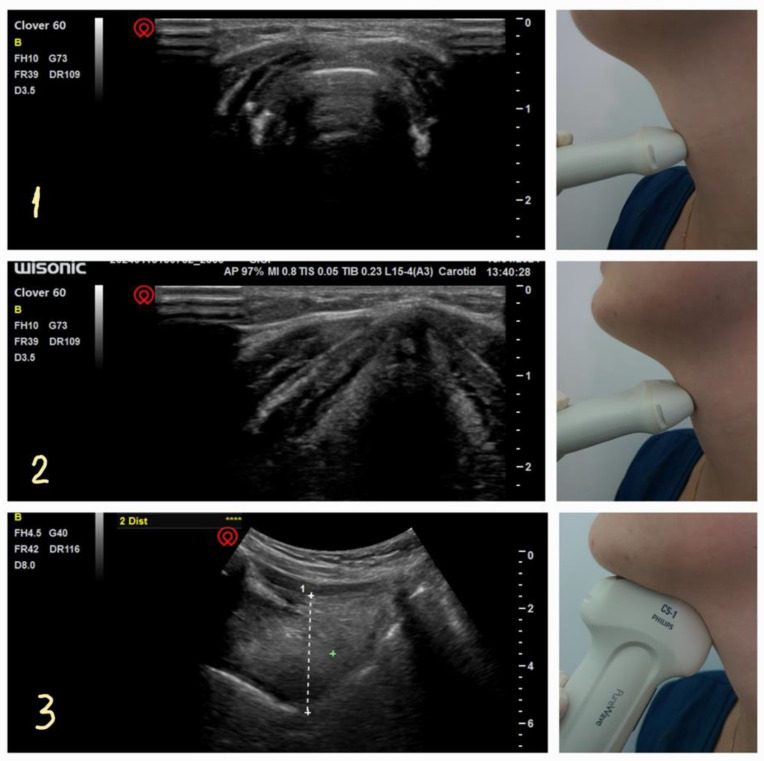
Ultrasonographic view of the cricothyroid membrane (**1**), glottis at the level of thyroid cartilage (**2**), and tongue base from sagittal suprahyoid view (**3**).

**Figure 6 diagnostics-14-00610-f006:**
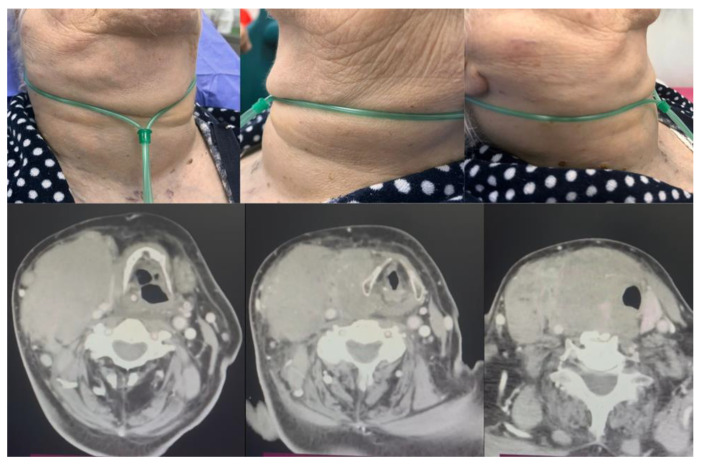
Comparative aspect between the clinical presentation and CT images of a massive thyroid tumor with compression effect on the airway.

**Figure 7 diagnostics-14-00610-f007:**
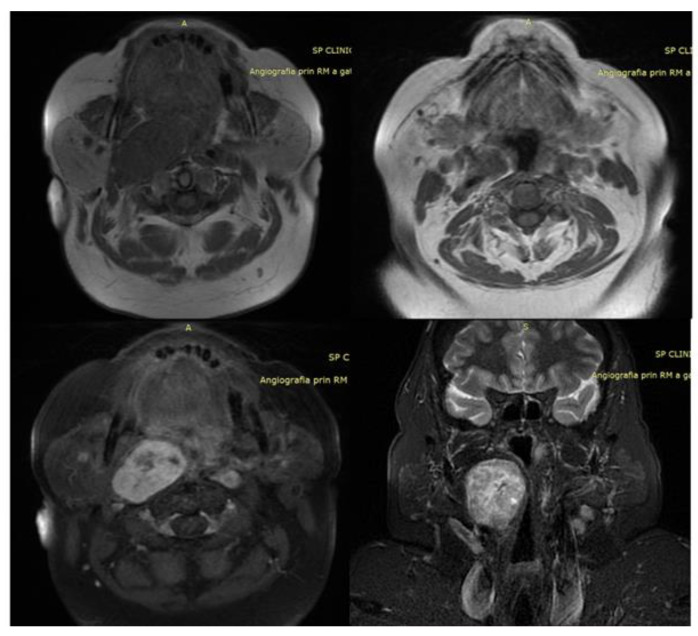
MRI images of a parapharyngeal tumor with displacement of the oropharynx and hypopharynx (vagal neurinoma).

**Figure 8 diagnostics-14-00610-f008:**
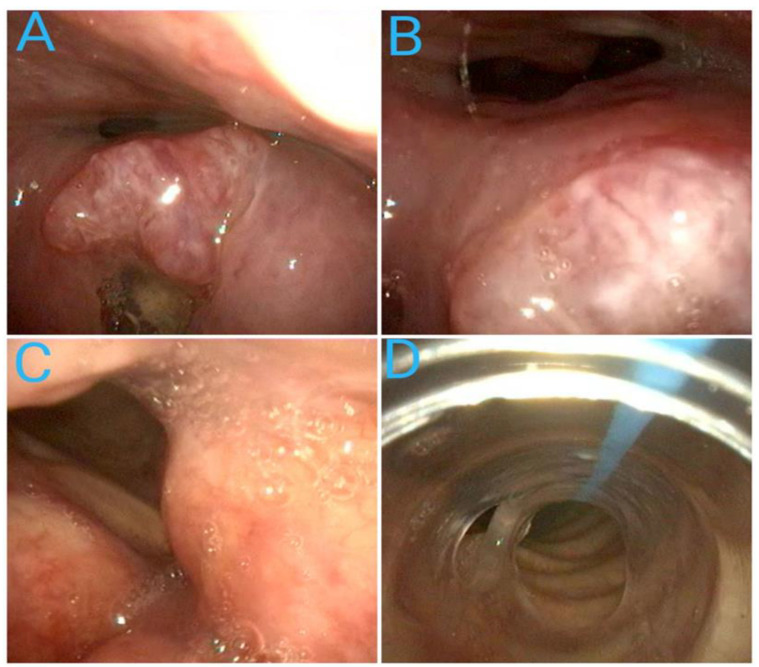
Awake intubation in a patient with a large epiglottis tumor. The sequence shows the advancement of the flexible scope armed with an intubating tube along the distorted anatomy at the level of the nasopharynx (**A**), epiglottis (**B**), and glottis (**C**) and the safe placement of the intubating tube inside the trachea (**D**) (Source: Ioan Florin Marchis—New methods of controlling the difficult airway in otorhinolaryngologic anesthesia—PhD Thesis. Cluj Napoca, 2022 [100]).

**Figure 9 diagnostics-14-00610-f009:**
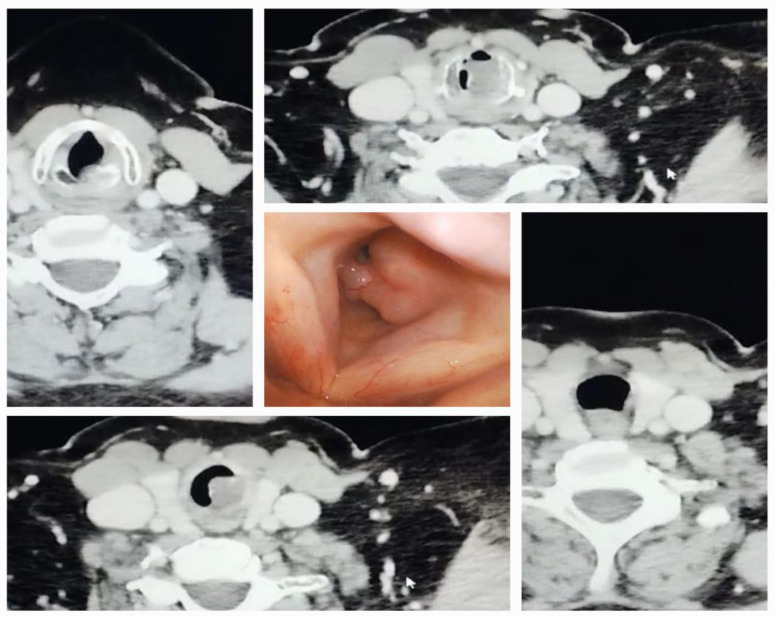
CT scan images and endoscopic view of a patient with stenotic subglottic tumor.

**Table 1 diagnostics-14-00610-t001:** Wilson score (range: 0–10).

Parameter	0 Points	1 Point	2 Points
Weight (kg)	<90	90–110	>110
Cervical spine mobility	>90	90	<90
Impaired jaw mobility	Interincisor gap ≥ 5 cm or able to protrude the lower teeth past the upper teeth	Interincisor gap < 5 cm and only able to protrude the lower teeth to meet the upper teeth	Interincisor gap < 5 cm and unable to protrude the lower teeth to meet the upper teeth
Retrognathia	Normal	Moderate	Severe
Prominent incisors	Normal	Moderate	Severe

**Table 2 diagnostics-14-00610-t002:** Predictors of difficult intubation and mask ventilation.

Predictors of Difficult Direct Laryngoscopy	Predictors of Difficult Face Mask Ventilation
Limited mouth opening	Higher body mass index or weight
Limited mandibular protrusion	Older age
Narrow dental arch	Male sex
Decreased thyromental distance	Limited mandibular protrusion
Modified Mallampati class 3 or 4	Decreased thyromental distance
Decreased submandibular compliance	Modified Mallampati class 3 or 4
Decreased sternomental distance	Beard
Limited head and upper neck extension	Lack of teeth
Cormack–Lehane Grade 3 or 4	History of snoring
Large tongue/epiglottis	History of neck radiation
Increased neck circumference	Obstructive sleep apnea

**Table 3 diagnostics-14-00610-t003:** Conditions associated with difficult airways.

Congenital	Acquired
Pierre Robin syndrome	Morbid obesity
Goldenhar syndrome	Pregnancy
Treacher Collins syndrome	Radiation of face or neck
Achondroplasia	Infections involving the airway (Ludwig’s angina, epiglottitis, abscess, papillomatosis)
Mucopolysaccharidoses	Allergic reactions (Quincke’s edema)
Micrognathia	Obstructive sleep apnea
Beckwith syndrome	Tumors involving the upper airway
Down syndrome	Trauma or burns of the head or neck
Cretinism	Ankylosing spondylitis
	Acromegaly

**Table 4 diagnostics-14-00610-t004:** Summary of studies regarding transnasal flexible fiberoptic laryngoscopy (TFL) and difficult airway prediction.

Authors, Year	N	Patient Population	Study Design	Main Findings
Barclay-Steuart et al., 2023 [29]	1099	Elective ENT surgery patients	Retrospective cohort	Lesions of the vestibular folds, supraglottic region, and arytenoids, viewing restriction of the rima glottidis covering ≥50% of the glottis area, and pharyngeal secretion retention were predictive of difficult airway management. When age, sex, BMI, and Mallampati scores were added, the AUC of the model reached 0.74. Lesions at the vocal cords, epiglottis, or hypopharynx were not predictive of difficult airway management. *
Sasu et al., 2023 [30]	252	Elective ENT and OMF surgery patients	Retrospective cohort	Vestibular fold lesions, epiglottic lesions, pharyngeal secretion retention, and restricted view of the rima glottis covering <50% and ≥50% were associated with a difficult videolaryngoscopy (alert issued by anesthetist after intubation).
Tasli et al., 2023 [39]	98	Rhinological and otologic surgery patients	Prospective cohort	The Tasli classification ** and Cormack–Lehane grade are moderately correlated (Pearson correlation coefficient = 0.582). A Tasli grade of ≥2b had a sensitivity of 73.8% and a specificity of 83.3% in predicting difficult intubation (defined as more than one attempt at successful tracheal intubation).
Rosenblatt et al., 2011 [34]	138	Elective surgery of the upper airway, ASA I-IV	Prospective cohort	When compared to a clinical evaluation of the airway alone, PEAE affected airway management plans for 26% of the patients (28 patients originally planned for AFOI underwent anesthetic induction and laryngoscopy, while 8 patients originally intended for intubation following induction underwent AFOI).

Abbreviations: N, sample size; ENT, otolaryngology; AUC, area under the Receiver Operating Characteristic (ROC) Curve; OMF, oromaxillofacial; ASA, American Society of Anesthesiologists; PEAE, Preoperative Endoscopic Airway Examination; AFOI, awake fiberoptic intubation; * Difficult airway management was defined as difficult intubation (≥2 attempts, conversion to a different intubation technique, or Cormack–Lehane grade III or IV) or difficult facemask ventilation. ** Tasli classification [40]: (1) glottis, arytenoids, and epiglottis visible; (2a) ≥50 of the glottis, entire arytenoids, and epiglottis visible; (2b) <50% of the glottis, entire arytenoids, and epiglottis visible; (3) glottis not visible, but entire arytenoids and epiglottis visible; (4) only the epiglottis is visible.

**Table 5 diagnostics-14-00610-t005:** Summary of relevant studies regarding ultrasonographic parameters and difficult airway prediction. Difficult laryngoscopy was defined as a Cormack–Lehane grade III or IV, unless otherwise specified.

Authors, Year	N	Patient Population	Study Design	Main Findings
Ning et al., 2023 [51]	502	Elective laparoscopic cholecystectomy	prospective observational	Mandible–hyoid bone angle < 125.5°, DGTC > 1.22 cm, and DSEM > 2.20 cm predicted DL with AUCs of 0.930, 0.722, and 0.702, respectively. Tongue width, cross-section area, and volume, mandible–hyoid distance and hyoid–glottis distance were not predictive of difficult laryngoscopy.
Bhagavan et al., 2023 [52]	96	Elective surgery, ASA I-II	prospective observational	DSHB > 0.66 cm and DSEM > 2.03 cm predicted DL with AUCs of 0.974 and 0.888, respectively.
Alessandri et al., 2019 [53]	194	Elective ENT surgery	prospective observational, single blinded	DSHB predicts DMV * (AUC = 0.929) and DL ** (AUC = 0.660). DSTI, DSEM, DSTJ, and DSAC also correlate with DMV and DL.
Petrisor et al., 2018 [54]	25	Elective surgery in morbidly obese patients	prospective observational	HMDR2 has the highest diagnostic accuracy for DL in morbidly obese patients, with a cut-off of 1.23 and an AUC of 0.92. HMDR1 and HMD in the ramped and maximally hyperextended positions were also predictive of difficult intubation, but HMD in neutral position was not.
Wu et al., 2014 [55]	203	Elective surgery	prospective observational	DSHB, DSEM, and DSAC predicted DL with cut-off values of 1.28 cm, 1.78 cm, and 1.1 cm and AUCs of 0.92, 0.90, and 0.85, respectively.
Adhikari et al., 2011 [56]	51	Elective surgery	prospective observational	DSHB and DSME predicted DL, while TT and anterior neck soft-tissue thickness at the level of the vocal cords, thyroid isthmus, and suprasternal notch did not.
Yao et al., 2017 [57]	2254	Elective surgery	prospective observational	TT > 6.1 cm and TT-to-TMD ratio > 0.87 predicted DTI and DL, with an AUC of 0.78 and 0.86 for DTI *** and 0.69 and 0.75 for DL.
Rana et al., 2018 [58]	120	Elective surgery, ASA I-II, non-obese	prospective observational	Pre-E/E-VC > 1.77 cm and HMDR1 < 1.085 predicted DL with AUCs of 0.868 and 0.871, respectively.
Falcetta et al., 2018 [46]	301	Elective surgery	prospective observational, single blinded	mDSE > 2.54 cm and PEA > 5.04 cm^2^ predicted DL ** with AUCs of 0.906 and 0.93, respectively.
Xu et al. 2022 [59]	1000	Elective surgery, ASA I-III	prospective case-cohort study	Ultrasound model consisting of 3 parameters: TT > 61 mm, mandibular condyle mobility ≤ 10 mm, and HMD ≤ 51 mm (1 point each). Scores > 1 point predicted DTI (Se = 85%; Sp = 81%; AUC = 0.89) and DL (Se = 75% and Sp = 82%; AUC = 0.84) ***.
Agarwal et al., 2021 [60]	1043	Elective surgery, ASA I-III	prospective, observational, double-blinded cohort trial	Ultrasound model consisting of 4 parameters (TT > 5.8 cm, DSHB > 1.4 cm, DSEM > 2.4 cm, and VH) predicted difficult intubation with an AUC = 0.992.
Yao et al., 2017 [61]	484	Elective surgery, ASA I-III	prospective observational	Mandibular condyle mobility ≤ 10 mm predicted DL (Se = 81%; Sp = 91%; AUC = 0.93).
Kaul et al., 2021 [62]	100	Elective surgery, ASA I-II	prospective observational	DSAC > 1.68 cm (Se = 100%; Sp = 95%; AUC = 0.999), DSEM > 1.34 cm (Se = 93%; Sp = 82%; AUC = 0.975) and DSHB > 0.98 cm (Se = 92%; Sp = 69%; AUC = 0.799) predicted DL.
Udayakumar et al., 2023 [63]	100	Elective surgery, ASA I-III	prospective observational	DSEM > 2.03 cm (Se = 97%; Sp = 79%; AUC = 0.91) and DSVC > 1.12 cm (Se = 80%; Sp = 88%; AUC = 0.84) predicted DL.
Lin et al., 2021 [64]	41	Elective surgery, ASA I-III	prospective observational	TT > 6.96 cm was not predictive of DTI but predicted DMV (Se = 50%; Sp = 87%; AUC = 0.72) *
Sotoodehnia et al., 2023 [65]	123	ED, requiring RSI, excluding neck or head trauma or those with airway obstruction	prospective observational	DSAC and DSTI predicted DL and DTI. HV predicted DTI but not DL. DSHB predicted DL but not DTI. DBAC predicted neither.
Li et al., 2023 [66]	151	Comatose patients undergoing emergency intubation	prospective observational	A sum of DSHB and DSAC >1.9 was predictive of DL, with an OR = 7.76.

Abbreviations: N, sample size; DGTC, distance from the glottis to the thyroid cartilage; DSEM, distance from the skin to the epiglottis midway between the hyoid bone and thyroid cartilage (thyrohyoid membrane); DL, difficult laryngoscopy; ASA, American Society of Anesthesiologists; AUC, area under the Receiver Operating Characteristic (ROC) Curve; DSHB, distance from the skin to the hyoid bone; ENT, otolaryngology; DMV, difficult mask ventilation; DSTI, distance from skin to thyroid isthmus; DSTJ, distance from skin to trachea at jugular notch; DSAC, distance from skin to anterior commissure of the vocal cords; HMDR2, hyomental distance ratio 2 (maximum hyperextension divided by neutral position); HMDR1, hyomental distance ratio 1 (ramped position divided by neutral position); HMD, hyomental distance; TT, tongue thickness; TMD, thyromental distance; DTI, difficult tracheal intubation; Pre-E/E-VC, ratio of pre-epiglottic space/distance between epiglottis to midpoint of vocal chords; mDSE, median distance from skin to epiglottis; PEA, pre-epiglottic area; Se, sensitivity; Sp, specificity; VH, invisibility of the hyoid bone (on sublingual ultrasound scan); ED, emergency department; RSI, rapid sequence induction and intubation; DBAC, distance between arytenoid cartilages; * difficult mask ventilation was defined as a Han score of III or IV [67]. ** difficult laryngoscopy was defined as a Cormack–Lehane grade 2B or higher. *** difficult tracheal intubation was defined as >2 attempts with conventional laryngoscopy, the need for an alternative technique, or an intubation attempt lasting >10 min.

**Table 6 diagnostics-14-00610-t006:** Summary of findings regarding cervical radiography and difficult airway prediction. Difficult laryngoscopy was defined as a Cormack–Lehane grade III or IV.

Authors, Year	N	Patient Population	Study Design	Main Findings
Khan et al., 2013 [72]	4500	Elective surgery, ASA I-III	Prospective	Radiological parameters (atlantooccipital gap, mandibular angle, mandibular depth, and mandibulohyoid distance) had a low predictive power for DL.
Oh et al., 2020 [73]	184	Cervical spine surgery patients intubated with an Optiscope video stylet and manual inline stabilization	Retrospective	No parameters measured on lateral cervical X-ray or RMN correlated with DTI (defined as failure on the first attempt or intubation time > 90 s)
Han et al., 2018 [74]	315	Cervical spine surgery patients	Retrospective	A vertical distance from the superior aspect of the hyoid bone to the mandibular body of ≥20 mm predicted DL. (Se = 77.8; Sp = 71.3; AUC = 0.832). Extension angle of A * ≥ 38° predicted DL. (Se = 74.1; Sp = 65.5; AUC = 0.802)
Gupta et al., 2010 [75]	157	15–65 years old, elective surgery	Prospective	Maxillo-pharyngeal angle on lateral cervical X-ray correlates with DL.
Kharrat et al., 2022 [76]	71	Elective transoral microsurgery for laryngeal tumors	Prospective	A longer maxilla and shorter atlantooccipital distance on lateral cervical X-ray were correlated with difficult laryngeal exposure (defined as laryngeal exposure limited to the posterior third of the vocal cords or less when suspension direct laryngoscopy was performed by ENT surgeons).
Zhou et al., 2021 [77]	270	Cervical spine surgery patients	Prospective	C2C6AR ** < 1 predicted difficult laryngoscopy with Se = 0.88, Sp = 0.3, and AUC = 0.714.
Liu et al., 2020 [78]	104	Cervical spine surgery patients	Retrospective	Angle E *** < 19.9° (Se = 0. 885; Sp = 0.910; AUC = 0.929), distance from hard palate to upper incisors > 30.1 mm (Se = 0.769; Sp = 0.769; AUC = 0.819) and atlantooccipital gap < 7.3 mm (Se = 0. 731; Sp = 0.564; AUC = 0.636) predicted need for assisted intubation techniques ****.

Abbreviations: N, sample size; DL, difficult laryngoscopy; DTI, difficult tracheal intubation; Se, sensitivity; Sp, specificity. AUC, area under the Receiver Operating Characteristic (ROC) Curve; ASA, American Society of Anesthesiologists; * extension angle A was defined as the angle between the line along the occlusal surface of the maxillary teeth and a line drawn from the upper incisors to the antero-inferior border of the C6 vertebra with the neck extended. ** C2C6AR was defined as the ratio of the angles formed by the lines drawn through the inferior extremities of the C2 vertebra and C6 vertebra in extended and neutral positions. *** angle E was defined as the angle between a line uniting the postero-superior point of the hard palate with the lowest point of the occipital bone and the line uniting the infero-anterior and infero-posterior points of the C2 vertebra. **** assisted intubation was defined as the need to use alternative airway devices after failure to intubate with a Macintosh laryngoscope within 3 attempts.

**Table 7 diagnostics-14-00610-t007:** Summary of relevant studies regarding computed tomography (CT) parameters and their predictive value for difficult airways. The outcome variable was a difficult laryngoscopy (defined as Cormack–Lehane grade III or IV), unless otherwise specified.

Authors, Year	N	Patient Population	Study Design	Parameters Measured	Se (%)	Sp (%)	AUC
Kim et al., 2021 [79]	281	Elective thyroidectomy patients for suspected malignancy, ASA I-III	Retrospective	dMV > 2.33 cm	75.0	93.8	0.884
				dME > 3.27 cm	87.5	48.6	0.691
				dSV > 3.37 cm	77.5	65.6	0.742
				dSE > 4.54 cm	80.0	44.4	0.635
Lee et al., 2019 [69]	90	Acromegaly patients undergoing transsphenoidal removal of pituitary tumor	Retrospective	Tongue area > 2600 mm^2^	76	61	0.68
Li * et al., 2022 [80]	24	≥14 years old, surgery for ankylosing spondylitis/idiopathic scoliosis	Retrospective Cohort	Pharyngeal volume < 16 mL	85.7	70.6	nr
Yasli * et al., 2023 [81]	60	ASA I-II patients undergoing bimaxillary orthognathic surgery	Prospective	Vertical diameter of INV region ≤ 1.09 cm	75	71.43	0.77
				Horizontal diameter of INV region ≤ 0.39 cm	68.75	85.71	0.773
Mao et al., 2019 [82]	69	Children with Robin sequence undergoing MDO	Retrospective	Airway cross-section area at epiglottis tip > 36.97 mm^2^	100	62.5	0.8125
Del Buono et al., 2018 [83]	37	Elective surgery ASA I-IV	Retrospective	Neck fat volume	ns	ns	ns

Abbreviations: N, sample size; Se, sensitivity; Sp, specificity; AUC, area under the Receiver Operating Characteristic (ROC) Curve; dMV, membrane-to-vallecula distance (measured from the middle of the thyrohyoid membrane to the closest concave point on the vallecula); dME, membrane-to-epiglottis distance (measured from the middle of the thyrohyoid membrane to the most distant point on the epiglottis); dSV, skin-to-vallecula distance (extended distance straight anterior to the skin from the dMV); dSE, skin-to-epiglottis distance (extended distance straight anterior to the skin from the dME); ASA, American Society of Anesthesiologists; nr, not reported; INV, internal nasal valve; ns, not significant; MDO, mandibular distraction osteogenesis; ns, not specified; * outcome was difficulty in guiding the endotracheal tube through the nasal passage.

## Data Availability

Not applicable.

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
