# Peer review of "Trends in Preoperative Airway Assessment"

_diagnostics, 2024, doi:10.3390/diagnostics14060610_

Round 1
Reviewer 1 Report
Comments and Suggestions for Authors
The authors, Marchis et al, presented a significant issue regarding the assessment of the bronchial tree from an anesthesiological perspective before planned intubation and operative treatment. The paper essentially covers a comprehensive spectrum of evaluation, thoroughly and critically assessing the available methods for its examination. In the presented report, the authors consider patients treated for head and neck tumors to be one of the most challenging problems, a viewpoint that is widely agreed upon. Another group not mentioned by the authors includes patients with post-intubation tracheal stenosis or those who have undergone tracheal resection. In this group, assessment through classical and rigid bronchoscopy takes priority before planned interventions. I suggest supplementing the presented report with the clinical aspect of this group. The authors have presented an interesting paper that meets the criteria for publication.
Comments on the Quality of English LanguageIn the case of the paper being accepted for publication, I would recommend submitting it for professional language editing.
Author Response
We are grateful for the comprehensive review. We agreed with the reviewer's point of view and modified the manuscript according to his request, adding information about the airway assessment of patients with stenotic tracheal pathology. We apologize for omitting this group. We highlighted the added text in yellow in the 3.1, 3.2, 3.3, and 3.5 subheadings and hope we have reached the reviewer's expectations. Since this subject is complex and involves a multidisciplinary, personalized approach, we outlined strictly the principles of airway assessment connected with this complex airway pathology.
We agree to improve the quality of the manuscript with professional language editing upon acceptance.
Reviewer 2 Report
Comments and Suggestions for Authors
The authors present a comprehensive and valuable narrative review of literature regarding airway assessment. The manuscript is well written, easy to read and brings a good insight into traditional and some of the new airway assessment possibilities. A valuable addition is the part about AI use in this area which has a potential for significant use in the future.
There are some minor issues I would like the authors to address:
1. In the Methods section, please add the time period when the literature was searched.
2. Please add a flowchart of literature search.
3. Please add subheading regarding Results.
4. Both physical examination and nasoendoscopy have the same subheading 3. Please correct.
Author Response
We are happy with the positive analysis and want to thank the reviewer. We addressed point by point the comments:
- We added the period when and since we searched the literature
- We added the flow chart as requested
- We modified the subheadings as requested, and we corrected the mistake when numbering the subheadings
Reviewer 3 Report
Comments and Suggestions for Authors
Dear Authors,
I find your work to be valid, well-structured, and well-written. The purposes of your narrative review are clear. The discussion summarizes the methods covered. The bibliography that inspired you is very extensive, and for this, I commend you. In my humble opinion, the work is suitable for publication. Just one note: in Table 6, the study design of Khan et al. is replaced with "Iran." Could you please correct this? Thank you.
Author Response
We are obliged for the reviewer's appreciation of our work. We modified Table 6 as requested and apologize for our error.